# Energy-Efficient Spatial Query-Centric Geographic Routing Protocol in Wireless Sensor Networks

**DOI:** 10.3390/s19102363

**Published:** 2019-05-22

**Authors:** Xing Wang, Xuejun Liu, Meizhen Wang, Yunfeng Nie, Yuxia Bian

**Affiliations:** 1Key Laboratory of Virtual Geographic Environment (Nanjing Normal University), Ministry of Education, Nanjing 210023, China; jwangxing0719@163.com (X.W.); wangmeizhen@njnu.edu.cn (M.W.); 2State Key Laboratory Cultivation Base of Geographical Environment Evolution (Jiangsu Province), Nanjing 210023, China; 3Jiangsu Center for Collaborative Innovation in Geographical Information Resource Development and Application, Nanjing 210023, China; 4School of Information and Engineering, Nanchang Hang Kong University, Nanchang 330000, China; Nieyunf@gmail.com; 5College of Resources and Environment, Chengdu University of Information Technology, Chengdu 610225, China; byx310@163.com

**Keywords:** wireless sensor network, geographic routing protocol, spatial query, energy balance

## Abstract

In data-centric wireless sensor networks (WSNs), sensing data have a high time–space correlation. Most queries are spatial and used to obtain data in a defined region. Geographic routing (GR) protocols are the optimal choice for routing spatial queries. However, several drawbacks still exist in GRs, and these the include premature death of nodes and communication latency, which result in reduced network life and query efficiency. A new clustering GR protocol called quadtree grid (QTGrid) was proposed in this study to save energy and improve spatial query efficiency. First, the monitoring area was logically divided into clusters by a quadtree structure, and each grid’s location was encoded to reduce the memory overhead. Second, cluster head (CH) nodes were selected based on several metrics, such as distance from the candidate node to the grid center and adjacent CHs and residual energy. Third, the next-hop routing node was selected depending on the residual energy of the candidate node and its distance to the sink node. Lastly, a lossless data aggregation algorithm and a flexible spatial query algorithm were adopted to reduce the transmission of redundant data and meet the application requirements, respectively. Simulation results showed that compared with three related protocols, QTGrid has lower energy consumption and higher spatial query efficiency and is more suitable for large-scale WSN spatial query application scenarios.

## 1. Introduction

Wireless sensor networks (WSNs) are data-centric networks, and sensing data have a high time–space correlation [1,2]. Spatial queries in WSNs are useful for obtaining data concerning events limited to a well-defined geographic region [3]. Routing protocols are responsible for ensuring that spatial query data packets are transmitted from a target region through multiple relay nodes to a destination without being dropped or/and compromised [4,5]. In most WSN applications, sensor nodes are powered by batteries, which are impossible to recharge or replace after deployment [1]. A node has three basic states, namely, sensing, data processing, and data communication. Transmitting or receiving packets during data communication consumes a considerable amount of energy. Hence, efficiency and energy conservation are key issues in the design of routing protocols.

Routing protocols in WSNs can be divided into flat-based, hierarchical-based/clustering, and location-based/geographic routing (GR) depending on the network structure. When the location of nodes and the target area are known, GR uses this information to create routes, thereby consuming less energy than protocols that do not employ location information [6]. Thus, these protocols are usually the best choice for spatial queries [4,7]. Classical GRs, such as greedy perimeter stateless routing [8], geographic energy-aware routing [9], and graph embedding [10], are attractive approaches for WSNs because no end-to-end route is established before data transmission and network scalability [11]. However, GR still has drawbacks, such as high latency, high collision, and energy hole, and these need to be addressed [12,13]. The energy-efficient spatial query-centric GR protocol has elicited considerable attention due to the energy constraints of nodes.

Flat-based routing [14,15] protocols are unsuitable for large-scale WSNs because of their single structure and limited extension. Clustering is a popular topology control method for data routing through multi-hop communication [16]. According to whether a cluster is divided evenly or not, clustering protocols can be divided [17,18,19,20] into even clustering (EC) and uneven clustering (UEC). Compared with EC, UEC [21,22,23] is unsuitable for spatial queries for the following reasons: (1) the clustering process consumes substantial energy, (2) the unequal size of clusters easily leads to uneven energy consumption of nodes in different clusters, and (3) irregularity in network topology between clusters usually makes routing increasingly complex. Meanwhile, EC divides the entire sensing network into multiple equal-sized clusters, and the energy consumption of intra-cluster nodes communicating with the cluster head (CH) is approximately similar. Therefore, the energy consumption of EC protocols is more balanced than that of UEC protocols, and the network topology is more regularized, which is more conducive for spatial queries.

In summary, EC geographic routing protocols are suitable for spatial queries in large-scale WSNs, such as geographic adaptive fidelity (GAF) [24], Grid [25], and GeoGrid [26]. GAF enables nodes in the grid to sleep through a distributed negotiation mechanism, thereby reducing the energy consumption of nodes. However, GAF selects CHs randomly, which inevitably wastes energy and leads to a network that is not uniformly distributed. Grid selects CH according to the degree close to the grid geometry center to improve transmission stability. However, when the number of nodes is small or the network topology changes quickly, network holes are likely to emerge and interrupt the transmission. Based on Grid, GeoGrid uses two geocast forwarding methods that can effectively prolong network life and reduce network traffic. However, its spatial query efficiency is low, and the energy utilization is not efficient enough.

In this study, an EC-based GR called quadtree grid (QTGrid) was proposed to achieve energy conservation and spatial query efficiency for large-scale WSNs. QTGrid was discussed from the aspects of cluster partitioning and encoding, CH election, routing strategy, and data aggregation. First, the monitoring area was logically divided into clusters by a quadtree structure, and each grid’s location was encoded to reduce the memory overhead. Second, CH nodes were selected based on several metrics, such as distance from a candidate node to the grid center and adjacent CHs and residual energy. Third, the next-hop routing node was selected depending on the residual energy of the candidate node and its distance to the sink node. Fourth, a lossless data aggregation algorithm and a flexible spatial query algorithm were adopted to reduce the transmission of redundant data and meet the application requirements, respectively. Lastly, QTGrid and several related methods were compared and analyzed through simulation experiments.

The rest of this paper is organized as follows. Section 2 discusses the existing work related to the proposed technique. Section 3 introduces the details of the newly proposed routing protocol. Section 4 evaluates the protocol through simulations, and Section 5 presents the conclusions.

## 2. Related Work

EC-based improved GRs have been developed for many specific sensor networks in recent years [27,28,29,30,31]. However, we only reviewed studies that are relevant to our proposed work, that is, those that address energy efficiency [31,32,33,34,35,36], load balancing [37,38,39], and spatial query issues [40,41,42,43]. We organized this review by discussing energy-efficiency and load-balancing EC algorithms, followed by several spatial query efficiency GR issues.

### 2.1. Energy Efficiency and Load Balancing

GeoGrid evenly divides a monitoring area into several same-sized grids, and a CH is selected in each grid. CHs are responsible for forwarding data packets. In accordance with the distance between the current and target grids, GeoGrid performs data packet transmission in a grid-by-grid manner through CHs. Once a node is selected as the CH, it continues to work until it moves out of the grid or until the remaining energy reaches the minimum survival threshold. Then, a CH is re-selected. Hence, the premature death of several nodes easily occurs, leading to an energy hole [36]. Moreover, the division of network topology is simple and not efficient for spatial queries.

A grid-based fault-tolerant clustering and routing algorithm (GFTCRA) was proposed in a previous study [37] to address the hot spot problem by addressing the failure of CHs. GFTCRA follows a distributed approach and distributed run time management is used for all nodes of any cluster in case their CHs fail. The simulation results of the study showed that GFTCRA outperforms two relevant algorithms in terms of tolerance of CH sudden failure and network lifetime. However, GFTCRA does not consider any delay/hop count for data delivery to the sink, and spatial query efficiency must be improved.

A cost-based energy-balanced clustering and routing algorithm (CEBCRA) was developed in another work [16] to reduce energy consumption and prolong the network lifetime. First, CEBCRA selects CHs in a distributed manner on the basis of residual energy and neighbor cardinality. Second, each non-CH sensor node (also called a cluster member node (CM)) selects a CH within its communication range on the basis of the cost value of the CHs. Third, CEBCRA uses single-hop communication within each cluster and performs multihop communication among the clusters in data routing. For inter-cluster routing, a CH measures the cost of each path from itself toward the base station while selecting other CHs as a relay node for data forwarding on such paths. However, the algorithm complexity of CEBCRA is high, and additional communication overhead with surrounding nodes is required.

Another study proposed grid-based clustering protocol (GCP) [38] to enhance the energy efficiency and prolong the lifetime of large-scale WSNs. GCP divides the sensing field into a series of logical grid cells to form a cluster. CHs are selected based on the residual energy and location information of nodes. However, the location information encoded in GCP occupies a large space. Moreover, the complex topology in each cluster is not conducive for spatial queries.

The energy-efficient grid-based routing algorithm (EEGBR) [39] was proposed to prolong the network lifetime. In EEGBR, each grid selects its grid coordinator in consideration of the leftover energy of the nodes within the grid and separates the nodes to the sink by utilizing fuzzy logic. Therefore, energy consumption is reduced at the sensor nodes. Given that information transmitted from the source to the sink is performed by the grid coordinator, which acts as a relay node, the energy consumption in the routing process is minimized and network lifetime is improved. Although the selection of relay nodes shares the tasks and energy consumption of CHs, it increases the cost of node management and algorithm complexity.

### 2.2. Spatial Query Efficiency

A geographic routing protocol based on clustering for WSNs (GRCS) [40] was proposed to enhance energy conservation and the degree of success to reach the destination. In GRCS, the network is organized into a set of clusters in which a CH is periodically selected for each cluster. An improved clustering mechanism is employed in the routing process to optimize the path to the destination dynamically. Furthermore, GRCS combines three routing strategies for data forwarding. However, CHs are selected based on the distance from the cluster to the destination, which may cause an energy hole.

Grid-based enabled geographic routing (GEGR) [41] was proposed to reduce the energy cost in uniformly deployed dense WSNs. Clusters are set up depending on the construction of a 2D logical grid in the geographical region. GEGR uses the CH in each grid to route data. This scheme limits the use of broadcasting in the WSN to the process of CH election and the process of constructing and maintaining the table of neighboring CHs in adjacent grid cells. However, in GECR, the load balance of sensor nodes and CH failures are not considered, and the scheme is unsuitable for large-scale WSNs.

Markov chain model-based optimal cluster heads (MOCHs) [42] use a simple strategy to select the optimal number of CHs and overcome the problem of uneven energy distribution in the network. In MOCHs, the base station (BS) controls the number of CHs, and the CHs control the cluster members in each cluster in such a restricted manner that a uniform and even load is ensured. However, the partition of clusters is irregular, which is not conducive for spatial queries, and the Markov chain algorithm is complex.

The quadtree-based data collection structure (QTBDC) [43] is a logical hierarchical cluster structure based on the quadtree structure. All sensor nodes are encoded, and a logical multi-level cluster is constructed. A lossless data monitoring mechanism is used to reduce the communication cost. Simulations have shown that QTBDC prolongs the life cycle of WSNs, and the monitoring mechanism may result in a massive reduction in data traffic. However, the node closest to the physical center of the grid easily causes the premature death of several other nodes, and the next-hop node selection approach may easily lead to an energy hole.

Another study proposed the grid-based clustering and combinational routing (GCCR) [20] algorithm based on the grid structure to improve the network lifetime and scalability of large-scale WSNs. A suitable grid size is calculated according to the size of the area and transmission range, and a virtual grid structure is constructed. A CH is selected in each grid on the basis of the nearest distance to the midpoint of the grid. A localized single-path strategy is employed to forward data within a grid. An angular inclination-based combinational routing model is implemented to forward aggregate data from the CH to the sink. However, the calculation of angular inclination increases computational complexity, and the scheduling of packets is disregarded.

A comparison of clustering GR protocols is shown in Table 1. We conclude that EEGBR, GRCS, and GEGR are unsuitable for large-scale network requirements. GeoGrid performs well in energy saving but poorly in load balancing and spatial query efficiency. The load balancing of QTBDC and the spatial query efficiency of GCCR should be improved.

## 3. QTGrid Routing Protocol

The proposed quadtree grid (QTGrid) routing protocol is introduced in five aspects, namely, cluster setup, the election of CH and parent CH (PCH), routing strategy, data aggregation, and spatial query.

### 3.1. Cluster Setup

Figure 1 shows the construction of grids. The monitoring area is initially expanded, and the smallest square G that completely covers the monitoring area and the sink node in the center of the network is defined.

Quadtree [44,45] is a data structure that has been widely used in data clustering, computational geometry, and image processing. Quadtree partitioning of a 2D area facilitates the hierarchical spatial indexing of individual sub-regions and provides easy access routes to such sub-regions [11,12]. This partitioning method was employed in QTGrid to divide the expanded monitoring area. By using the basic idea of the quadtree management scheme, four adjacent level-1 grids formed quad region G. In each level-1 quad region, four adjacent non-overlapping single quads formed a nested level-2 quad. On the basis of this rule, each level-3 quad was also subdivided and so forth. In this manner, the expanded monitoring area G was recursively divided into logical quads.

As shown in Figure 1, the abovementioned operations were recursive until the entire G is divided into $4^{Z}$(Z≥0) sub-zones, where Z is the quadtree depth determined by specific applications (e.g., the spatial resolution of sensor data). Assuming that the side length of G was L and the side length of the level-Z grid was l, Equation (1) is followed.

(1)$$l=L/2^{Z}$$

#### 3.1.1. Grid Position Encoding

In QTGrid, the pre-encoding of sub-regions at each level can involve stacks of area identification codes. The identification codes of level-1 grids were expressed as 01 (northwest, NW), 11 (northeast, NE), 00 (southwest, SW), and 01 (southeast, SE), and the identification codes of level-2 grids can be expressed as 0100, 0111, and so on.

The shaded portion grid X in Figure 1 is 010110, which was taken as an example. The first two bits “01” represent the upper left quarter level-1 sub-zone of the entire G. The middle two bits “01” represent the upper left quarter level-2 sub-zone, and the last two bits “10” represent the upper right quarter level-3 sub-zone.

QTGrid encodes each grid using the Morton code [43,46] to convert the 2D coordinate into an M number. The M coding method is not explained in detail in this paper, but the M code of the entire monitoring area is shown in Figure 2. Combining Figure 1 and Figure 2, the shaded portion in Figure 1 (010110) can be converted into M code (22) to represent its location. After encoding, the M code of a grid and the intra-cluster ID ($N_{\mathrm{id}}$) of a sensor node form the sensor node’s identification code format, which is $\left(M,N_{\mathrm{id}}\right)$.

Numerous GRs use the 2D coordinate (m,n) to identify the location of a grid, where m is the row number and n is the column number. If m,n are represented by double-precision values, then 8 bytes of space is consumed. In contrast, QTGrid uses M code, which only needs to occupy 2 bytes, thereby reducing the data size. Moreover, each node can deduce its neighbor CHs and other levels’ CHs by using the M code without communication and election between nodes (details in Section 3.2). This method decreased the data transmission energy consumption while recording the hierarchical structure of the network and the topological relationship between nodes.

#### 3.1.2. Communication Radius

Assuming that the transmitted power of a sensor node is adjustable, the most extreme case in intra-cluster communication is that a CH and its CM are located at both ends of the diagonal line of the grid, as shown by the dotted line pointing from A to O in Figure 3. To ensure normal communication between CH A and CM O, the intra-cluster nodes’ communication radius R and the side length l of the grid should satisfy $R\geq \sqrt{2}l$. In the communication between two adjacent clusters, the most extreme case is represented by the solid line pointing from CH A to CH I in Figure 3. The CH communication radius should meet $R\geq 2\sqrt{2}l$ to ensure normal communication between CH A and CH I, which is also confirmed in the literature [11]. Therefore, in our simulation setup (Section 4.1), we set the communication range of CHs is equal to $2\sqrt{2}$ times the side length.

### 3.2. CH and PCH Election

#### 3.2.1. CH Election

CH election is crucial for GRs. In the initial stage, QTGrid selects the node closest to the center of the grid as the CH [11,26]. Then, in the re-selection of CH, QTGrid comprehensively considers energy consumption and load balancing. The CH nodes were selected based on metrics, such as distance from the candidate node to the grid center and the eight adjacent CHs and residual energy. QTGrid defines chs as the coefficient of CH election and selects the node with the minimum chs as the CH. chs can be calculated with Equation (2).
(2)$$chs=\alpha (1-\frac{E_{current\;i}\;}{E_{initial\;}{}_{i}})+\beta \frac{\sqrt{2}d}{l}+\gamma \frac{\sum_{i=1}^{8}D_{i}}{8\times 2\sqrt{2}l}$$
where α, β, and γ are equilibrium coefficients; a>0, β>0, γ>0 and they meet α+β+γ=1; $E_{current\;i}$ and $E_{initial\;i}$ are the current residual energy and initial energy, respectively; $D_{i}$ is the sum of the distance between the nodes to the eight adjacent CHs; d is the distance between the node and grid center; l is the side length of the current level grid.

QTGrid periodically broadcasts the CH election message to all nodes in the network at an interval of T seconds. After receiving the message, each cluster starts a new round of CH election. When a node in the cluster was elected as the CH by Equation (2), the election information was broadcasted and recorded by the CMs and adjacent clusters. In this manner, CHs with poor performance were replaced by more suitable CMs to avoid the premature death of several CHs and balance the energy loss of the intra-cluster nodes.

#### 3.2.2. PCH Election

An effective network topology control structure can improve the efficiency of network communication protocols while extending the network lifetime [47]. After quadtree hierarchical partitioning of the network, QTGrid adds management nodes at the corresponding quadtree level. In addition to the sink node, each cluster level sets a PCH as the management node responsible for collecting information on the four lower sub-CHs (SCH), aggregating the collected data, and transmitting them to the PCH. To reduce computing expenses of the PCH selection algorithm, QTGrid selects the SCH closest to the sink node as its PCH. The Algorithm 1 is the pseudo-code of PCH election. In this algorithm, a CH can calculate the M code of the cluster, where its PCH is located only based on its own identification code. Therefore, no extra communication consultation is required.

**Algorithm 1** PCH Election m ← *the M code of the CH**Level i ← the level of the PCH* *d ← the depth of the quadtree* *sink ← the location of Sink node* *Get PCH (*m*, level i, d, sink)* *{* *begin*  *[CHs] = [NW, NE, SW, SE]// Get the 4 CHs of the i-th level whose M code are*
m *[Xc,Yc] = (M,i)// Get the center coordinate of the area where the M code is*
m  *[Xs,Ys] = sink //Get the coordinates of the sink node*  *If(Xs>Xc)&(Ys>Yc)*   *PCH =SE*  *Else if(Xs<Xc)&(Ys>Yc)*   *PCH = SW*  *Else if(Xs<Xc)&(Ys>Yc)*   *PCH = NW*  *Else (Xs>Xc)&(Ys<Yc)*   *PCH = NE*
  *end* *end* *Return PCH// Return the calculated PCH node* }

### 3.3. Routing Strategy

The topology structure of the network is shown in Figure 4, where Z is the depth of the quadtree. Hence, QTGrid is a multilevel clustering network structure, in which the data routing process adopts the transmitting-while-aggregating method (details in Section 3.4).

QTGrid was performed in a grid-by-grid manner by using level-Z CHs to transmit data packets. The sensor nodes collect and forward the data to the level-Z PCHs. The level-Z PCHs fused the data and forward them to the upper level-(Z-1) PCH. In this manner, sensing data were collected and sent to the PCHs of the next level then aggregated and forwarded to the higher-level PCH until they were received by the sink node. Therefore, next-hop node selection for data forwarding is a key issue in the routing strategy.

To solve the premature death of management nodes (CHs) and the energy hole problem, QTGrid defines NHop to select the next-hop routing node.

As shown in Figure 5, we assume that D $\left(x_{1},y_{1}\right)$ is a level-2 PCH, A, B, and C are the SCH of D, J $\left(x_{0},y_{0}\right)$ is the level-3 PCH, and E $\left(x_{2},y_{2}\right)$, H $\left(x_{3},y_{3}\right)$ and I $\left(x_{4},y_{4}\right)$ are D’s adjacent level-2 PCHs. Assuming that D needs to send data to its PCH J, eight adjacent peer CHs were available in each grid for selection. To ensure a relatively short path, we selected three CHs whose grid centers were close to the Sink node as candidate next-hop routing nodes.

$l_{i}$ (where 1≤i≤3) is the distance from D to its i-th adjacent peer CH, $s_{i}$ is the distance from the i-th adjacent peer CH to J, and $L_{i}$ is the distance from D to J via the i-th adjacent CH. Then, we obtained
(3)$$L_{i}=l_{i}+s_{i}$$
where $l_{i}=\sqrt{{\left(x_{i}-x_{1}\right)}^{2}+{\left(y_{i}-y_{1}\right)}^{2}}$, $s_{i}=\sqrt{{\left(x_{0}-x_{i}\right)}^{2}+{\left(y_{0}-y_{i}\right)}^{2}}$. Therefore, $L_{i}$ can be rewritten as

(4)$$L_{i}=\sum_{j=0}^{1}\sqrt{{\left(x_{i}-x_{j}\right)}^{2}+{\left(y_{i}-y_{j}\right)}^{2}}$$

By comprehensively considering the distance and residual energy when selecting the next-hop routing node, QTGrid defines NHop as an evaluation factor for the candidate CHs, and the i-th candidate CH’s $NHop_{i}$ can be obtained by
(5)$$NHop_{i}=\lambda \frac{L_{i}}{L_{\max}}+\mu (1-p_{i})$$
where λ and μ are balance coefficients (where λ+μ=1, λ>0 and μ>0); $L_{max}$ is the maximum among $L_{2}$, $L_{3}$, and $L_{4}$; $p_{i}=E_{cuurent\;i}/E_{initial\;i}$ ($E_{current}{\;}_{i}$ and $E_{initial}{\;}_{i}$ are the residual and initial energy of the CH, respectively).

In reference to Figure 5, the routing strategy steps were as follows:Step 1.The sink node sent the data collection request to the event area, and the nodes in the routing area added their own energy information to the data package.Step 2.After receiving the package, the sensor nodes in the event area sent the data to the CH. Then, CHs A, B, and C sent the data to PCH D.Step 3.D aggregated the data and calculated the NHop values of the candidate next-hop routing nodes (E, H, I) using Equation (5).Step 4.D sent the data to the node with the smallest $NHop_{i}$ value. Here, we assumed that the NHop value of E was the smallest. Then, E continued to aggregate and forward the data to its next-hop routing node with the smallest NHop value until J received the data and forwarded them to the sink. Finally, data routing was completed.

In this demo, the routing path was D → E → F → J → sink.

### 3.4. Data Aggregation

Data aggregation is an important mechanism for achieving energy efficiency in WSNs. Many data aggregation protocols have been developed based on various techniques of optimizing delay and energy [48]. QTGrid employs a simple and efficient lossless data aggregation method to ensure data integrity.

During the QTGrid routing protocol, CHs at each level was regarded as an intermediate node in data aggregation. Sensing data were forwarded by strictly following the hierarchical structure of the quadtree, and data aggregation was performed simultaneously with forwarding. According to the routing hierarchy of the network, the CHs collected the sensing data of their SCHs and ranked them in ascending order according to the M codes. When the attribute values of adjacent M codes were equal, they were merged, and only the tuple with the smallest M code value was reserved. Figure 6 depicts a three-level WSN. Each level-3 CH sent the data to the level-2 PCH, and the level-2 PCH aggregated the data and sendt them to the level-1 PCH. The level-1 PCH aggregated the data and forwarded them to the sink node. Therefore, the data received by the sink node were the source data of the smaller space sent by the aggregated level-3 CHs. The literature [43] used a similar lossless aggregation method to collect the details of network sensing data and reduce the communication cost considerably.

### 3.5. Spatial Query

The spatial query of QTGrid has two main processes: sending a spatial query request and sensor data feedback. Window queries are the most common type of spatial queries [7]. In QTGrid, the spatial query request was sent to the defined region of interest (called window) and asked for data collected by the sensor nodes inside this region. On the basis of the advantage of the multi-level clustering network structure, the spatial query was transmitted through multi-level CHs. The grid level of the queried area can be customized by the user, which makes the spatial query flexible and suitable for practical application needs. Combining Figure 7, the two algorithms are given below. The Algorithm 2 is the pseudo-code of spatial query request sent, and the Algorithm 3 is the pseudo-code of sensor data feedback.

**Algorithm 2 Spatial Query Request Sent**
G ←the minimum rectangle covers the query windowR ←the minimum rectangle that totally covers the sink node and query window Gid ← the identification of the data packetX ← the sensor node that receives the packetsReceiveQueryRequestPackage (G, R, id){ Begin  if (X is within the zone R)   then begin    if (X is a CH)     then begin    //Judge that whether X has ever received the     //current request packet before     if (package ID==id)  then begin      Step1. X directly discards the packet;     end     else beginStep2. X sends the packet to its next routing hop CH;     endelse begin     Step3. X sends the packet to its CHend  end  else if (X is within the query window G)   then begin    if (X is a CH)     then begin     //Judge that whether X has ever received the     //current request packet before     if (package ID==id)  then begin       Step1. X directly discards the packet;     end     else begin      Step4. X sends the packet to its CMs;     endendelse begin      Step2. X sends the packet to its CHend   else begin   Step1. X directly discards the packet;   end  }

**Algorithm 3 Sensor Data Feedback**
X ← a sensor node in the query windowqueryWindow ← the query windowL ← the hierarchical level of X’s PCH in the query window input by the user Depth ← the depth of the quadtree structureGetDataOfRegion(queryWindow, L, Depth){Beginif (X is a CH) if (0<Level<Depth-1)then beginStep1. Collects and aggregates the required sensing data from CMs;Step2. Aggregates the required sensing data by the algorithm in Section 3.4;Step3. Uses PCH selection algorithm to calculate the level = L PCH;Step4. Calculates the routing to the level = L PCH;Step5. Sends the data to higher level PCH by formula (4)&(5) until level = L PCH receives the data packet;endif (Level =Depth-1)    then beginStep2. Aggregates the required sensing data by the algorithm in Section 3.4;Step6. Directly send packets to Sink;    endendelse beginStep7. X sends the sensing data to its CH    end    end}

## 4. Simulations and Analyses

### 4.1. Simulation Setup

Simulation was performed using Network Simulator MATLAB 7. The energy model is required to calculate energy consumption. In this study, the energy consumption of the node consisted of communication and calculation. Here, the communication energy model from the literature [49], which is commonly used, is introduced. The power consumption to transmit a *k*-bit message over distance d is given as follows:(6)$$E_{Tx}(k,d)=\{\begin{array}{c} E_{elec}k+k\varepsilon_{fs}d^{2}{,}_{{}_{}}d<d_{0}\\ E_{elec}k+k\varepsilon_{mp}d^{4}{,}_{}d\geq d_{0} \end{array}$$and the power consumption to receive the message is
(7)$$E_{Rx}=E_{Rx}(k,d)=kE_{elec},$$
where $E_{elec}$ denotes electronic energy, which depends on factors, such as the digital coding, modulation, filtering, and spreading of the signal. $\varepsilon_{fs}d^{2}$ or $\varepsilon_{mp}d^{4}$ depends on the distance to the receiver and the acceptable bit-error rate. $d_{0}$ is the threshold for switching to a different model, whose value can be set as adopted from the literature [50]. The calculation of nodes mainly included routing path and data aggregation, and the calculation energy consumption model is given as follows:(8)$$E_{C}=E_{r}+E_{da},$$where $E_{r}$ is the computation cost of routing per time and $E_{\mathrm{da}}$ is the computation cost of data aggregation.

QTGrid was designed for large-scale WSNs. According to the comparison results of various relevant GRs in Table 1, we only selected GRs suitable for large-scale networks for comparison. In view of the superior performance of GeoGrid [26], QTBDC [43], and GCCR [20] in energy saving, load balance, and spatial query efficiency, they were used for comparison with QTGrid in terms of average network energy consumption, number of network surviving nodes, and spatial query energy consumption. We set the network layer number of QTBDC and QTGrid to three, and the corresponding number of grids for setting GeoGrid and GCCR was 64. Other simulation parameters are shown in Table 2.

In all scenarios, the sink node was located at the center of the network and other nodes were randomly deployed. For illustrating the performance of QTGrid, we compared the network lifetime and spatial query of different protocols by calculating the average value of 100 trails with different random seeds in the following Figure 8, Figure 9, Figure 10 and Figure 11 (the confidence interval range with 95 percent of target values). Unless the sense data from the query area could not be transmitted to the sink node, we considered the network fail and stopped the simulation.

### 4.2. Network Lifetime

Network lifetime is defined as the average energy consumption of the network and the number of nodes surviving. The entire region was evenly divided into four parts, and the completion of spatial queries for each sub-region is regarded as a cycle.

Comparisons of QTGrid with three other protocols in terms of average energy consumption for target areas with 200, 400, and 800 sensor nodes are shown in Figure 8a–c, respectively. All the protocols reselected their CHs every 10 rounds.

The QTGrid protocol improved the CH selection method of DTBDC to balance the CHs’ energy consumption with the other intra-cluster nodes. Compared with GCCR, QTGrid reduced the frequency and distance of communication between nodes in the cluster to save energy based on the structural advantages of the quadtree. From Figure 8, we can observe the following:(1)Among the four routing protocols, QTGrid had the lowest average energy consumption, whereas GeoGrid had the highest one. Specifically, the average energy consumption of QTGrid was about 0.043 (200 nodes), 0.051 (400 nodes), and 0.075 J/round (800 nodes) lower than GeoGrid; about 0.032 (200 nodes), 0.043 (400 nodes), and 0.058 J/round (800 nodes) lower than GCCR; about 0.038 (200 nodes), 0.036 (400 nodes), and 0.049 J/round (800 nodes) lower than QTBDC.(2)When the number of nodes was 200, the average energy consumption of GCCR was about 0.006 J/round lower than that of QTBDC, but when the number of nodes was 400 and 800, the average energy consumption of GCCR was about 0.007 and 0.009 J/round higher than that of QTBDC, respectively. Compared with GCCR, QTDBC reduced the maximum length of data transmission and increased the number of forwarding hops to reduce the transmission duration. In the case of a network with few nodes (200 nodes), candidates for the next hop routing nodes tended to be few and resulted in routing paths that were not the shortest, which consumed much node energy to some extent. The above problem changed when many nodes were available in the network (400 and 800 nodes).

A comparison of the number of surviving nodes is shown in Figure 9a–c.

Figure 9 indicates that under the same constraints, GeoGrid had the fewest surviving nodes. QTGrid had more surviving sensor nodes and achieved a longer network lifetime than the three other protocols. When the number of nodes increased, the node death rate became slower in the QTGrid algorithm. This was because when designing the QTGrid, the energy balance of nodes in the network was concerned. To avoid the premature death of nodes with less residual energy, the in-cluster nodes with better conditions were selected as CHs. Therefore, the more nodes there were in the network meant that the energy consumption of the network was balanced to more candidate nodes in the same cycle, thus prolonging the network life. When the total number of nodes was 200, as shown in Figure 9a, GCCR had more surviving nodes than QTBDC, but when the total number of nodes was 400 and 800, as shown in Figure 9b,c, the result was reversed.

Therefore, as far as network life is concerned, QTGrid had the longest and GeoGrid had the shortest. Compared with GCCR, QTBDC was more suitable for energy saving of nodes in a large-scale network.

### 4.3. Spatial Query

In this section, we selected energy consumption and network transmission delay as metrics for the spatial query of the four routing protocols. All four protocols transmitted data in a grid-by-grid manner through CHs. The size of the grid was evenly divided, and CH was generally close to the center of the grid. It can be considered that the distances between CHs were almost the same. Therefore, we considered the number of transmission hops as the length of the transmission path.

The number of data transmission hops was of great significance to multi-hop WSNs. Usually, the shorter data transmission path (the fewer forwarding hops), the higher success rate of data forwarding [7]. In addition, increasing the number of data forwarding hops had a direct impact on network latency. Since the delay of data transmission was mainly caused by problems such as radio link contention, message processing time, and message queuing in per hop. The more forwarding hops, the greater network delay, and the more times of data receipt and transmission correspondingly, that is to say, the faster energy consumption. In summary, we simulated network latency by comparing the average number of routing hops.

The energy consumption of a spatial query used the average spatial query energy consumption ($\bar{E}$) of the monitoring area and the average number of routing hops for forwarding data as metrics. $\bar{E}$ is given as follows:(9)$$\bar{E}=\frac{\sum_{n=1}^{N}E_{Txi}+\sum_{n=1}^{N}E_{Rxi}+E_{C}}{N},$$where N is the number of nodes, $E_{Tx}$ is the power consumption to transmit the message, $E_{Rx}$ is the power consumption to receive the message, and $E_{C}$ is the power consumption for calculation.

Generating the network, we set up five different query windows, and slid the query window along the edge of the network until it covered the network completely. A query for each window was regarded as a round, and CHs were reselected every 10 rounds. Comparisons of QTGrid with three other algorithms in terms of $\bar{E}$ for the network randomly distributed with 200, 400, and 800 sensor nodes are shown in Figure 10a–c, respectively. The network latency corresponding to each scenario was compared, as shown in Figure 11a–c.

From Figure 10, we can observe the following:
(1)The three protocols improved $\bar{E}$ to varying degrees compared with GeoGrid, except for the network with 200 nodes where only one grid was covered. The reason was that compared with GeoGrid, the three other algorithms needed to consume more energy in CH election and replacement to obtain more energy-saving routes. In the above network, few candidate nodes were available for updating CHs, and the next-hop routing node and routing lines were not much different. However, excessive CH replacement increased the communication energy cost.(2)The $\bar{E}$ value of QTGrid was the lowest among those of the protocols, and the difference became more obvious with the increase in the number of nodes in the network. The $\bar{E}$ value of QTGrid was lower than that of GeoGrid and GCCR because of the use of data fusion technology, which can largely decrease the achieved redundant sensor data and the corresponding energy consumption of redundant data transmission. The $\bar{E}$ value of QTGrid was lower than that of QTBDC because the use of the new CH selection mechanism and new next-hop selection procedure made the nodes’ load more balanced.(3)When query windows had the same size, more nodes were available, and the superiority of QTGrid over the three other protocols in energy saving became more obvious because redundant data increased as the number of nodes increased. QTGrid fused sensing data, which played a crucial role in energy conservation. Although QTBDC and QTGrid used a similar data fusion strategy, QTGrid employed a more energy-efficient next-hop routing node selection method, which made the energy distribution of intra-cluster and inter-cluster communication balanced and energy conservation obvious for large-scale networks.

To measure the delay of data transmission, we calculated the average routing hops for the first 1000 rounds of the four protocols that were randomly distributed in the networks with 200, 400, and 800 sensor nodes. The results are shown in Figure 11a–c, respectively.

In Figure 11, the average number of routing hops of QTGrid is considerably smaller than that of GeoGrid and almost equal to that of GCCR and QTBDC. GeoGrid used location information to restrict the data propagation area. As the network ran periodically, the optimal routing nodes were more likely to die prematurely. Hence, GeoGrid had to choose the far-distance node as the next-hop routing node, which resulted in the largest average number of routing hops among the four protocols.

For these four protocols, the network structure of QTGrid and QTBDC was multi-level, while GCCR and GeoGrid were not. In general, when the query window was 1/64, 9/64 and 16/64 of the entire network, the average number of routing hops of QTGrid and QTBDC were smaller than that of GCCR and GeoGrid. The reason was that when the sink node was not within the query area, QTGrid and QTBDC transmitted the sensing data to the high-level CH (PCH), and this transmission was mostly short-distance path (fewer hops) communication. Next, the data of the entire query area were collected and transmitted to the sink node with long distance path communication at one time. While the transmission of the sensing data, always belonging to long-distance communication, was from CH of each grid to the sink node directly in GCCR and GeoGrid, the resulting average number of routing hops was larger than QTGrid and QTBDC.

Particularly, when the query window was 36/64 of the entire network, the sink node was within the query area, which was a certain level PCH in QTGrid and QTBDC. GCCR had a lower average number of routing hops than that of QTBDC and transmitted the data to sink directly in a grid-by-grid manner. Although GCCR adopted two kinds of data routing strategies (multi-hop and combinational routing), its number of average routing hops was slightly higher than that of QTGrid. By adopting the similar cluster partition approach, the improvement of the next hop routing node selection algorithms in QTGrid made the number of average routing hops smaller than that of QTBDC.

## 5. Conclusions

An energy-efficient and spatial query-centric clustering GR protocol called QTGrid was proposed. First, the monitoring area was divided logically into clusters by a quadtree structure, and each grid’s location was encoded to reduce the memory overhead. Second, CH nodes were selected based on several metrics, such as distance from the candidate node to the grid center and adjacent CHs and residual energy. Third, the next-hop routing node was selected depending on the residual energy of the candidate node and its distance to the sink node. Lastly, a lossless data aggregation algorithm and a flexible spatial query algorithm were adopted to reduce the transmission of redundant data and meet the application requirements, respectively. Extensive simulations were conducted on QTGrid and three related protocols (GeoGrid, QTBDC, and GCCR) in terms of performance in network lifetime and spatial query. The simulation results revealed that compared with the three other protocols, QTGrid had a longer network lifetime and higher spatial query efficiency and was thus more suitable for large-scale WSN spatial query application scenarios.

## Figures and Tables

**Figure 1 sensors-19-02363-f001:**
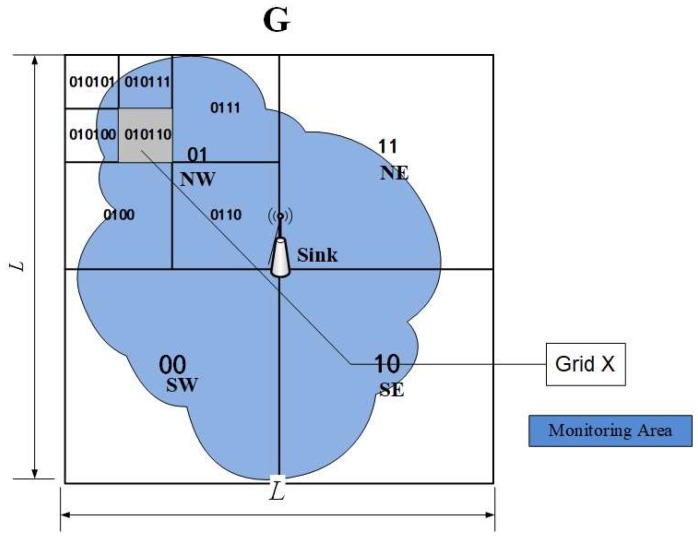
Construction of grids.

**Figure 2 sensors-19-02363-f002:**
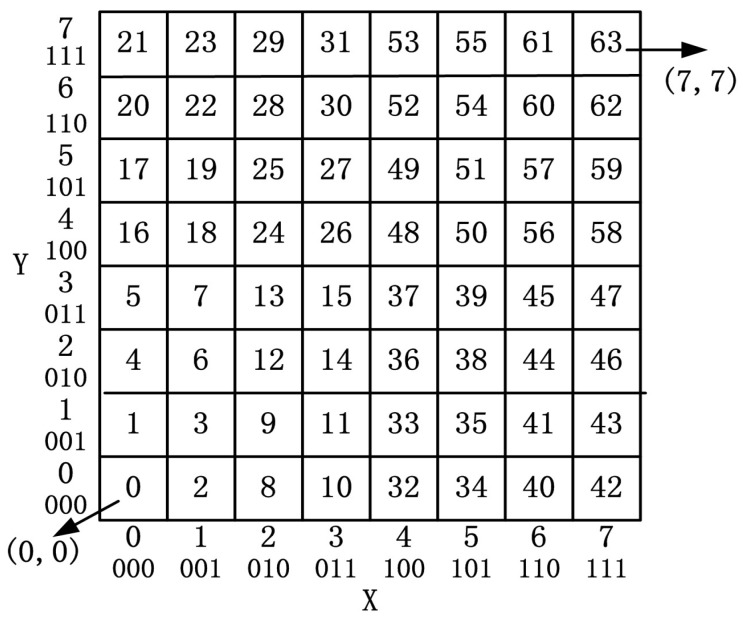
The M code of the expanding monitoring area.

**Figure 3 sensors-19-02363-f003:**
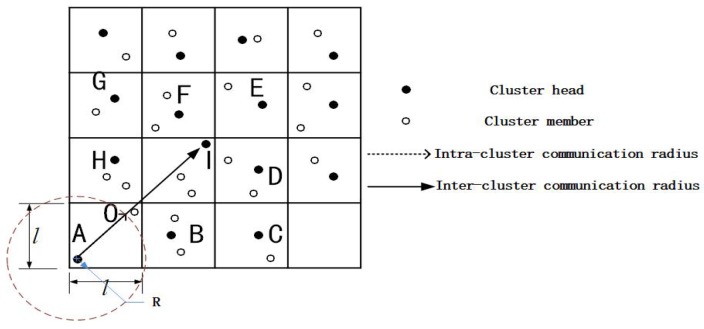
Communication radius of the node.

**Figure 4 sensors-19-02363-f004:**
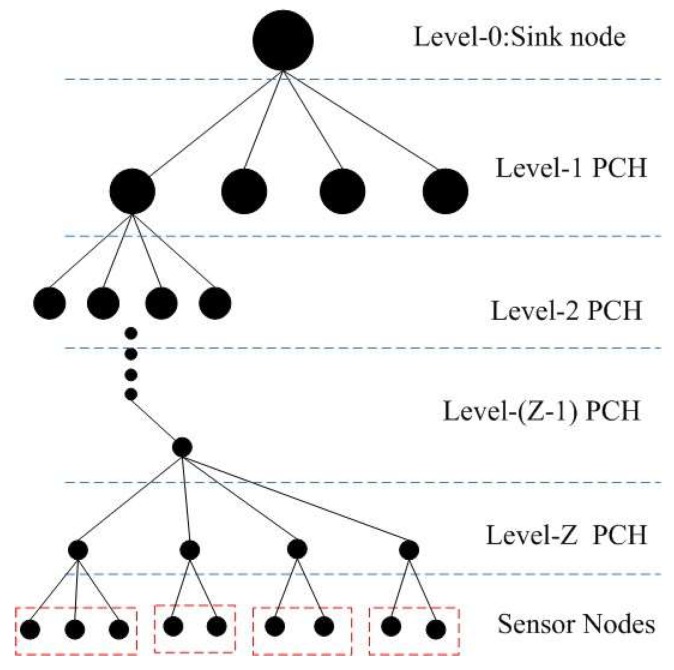
Topology structure of the network.

**Figure 5 sensors-19-02363-f005:**
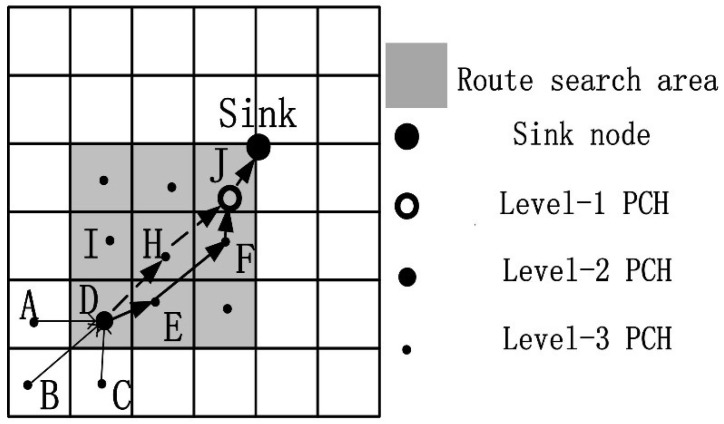
Routing strategies of QTGrid.

**Figure 6 sensors-19-02363-f006:**
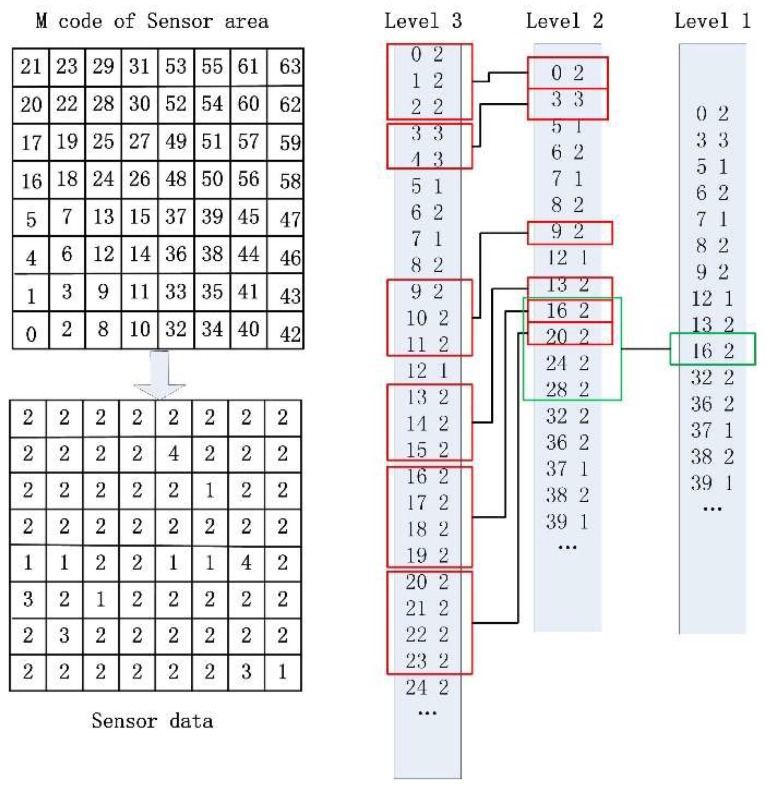
Data aggregation.

**Figure 7 sensors-19-02363-f007:**
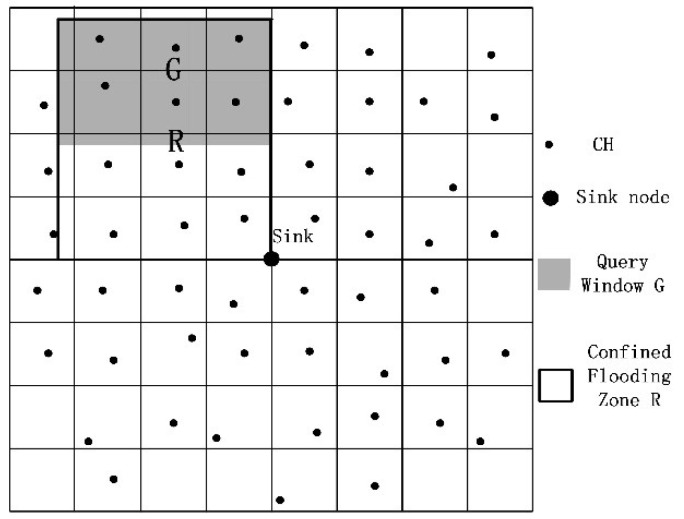
Sensor data transmission zone.

**Figure 8 sensors-19-02363-f008:**
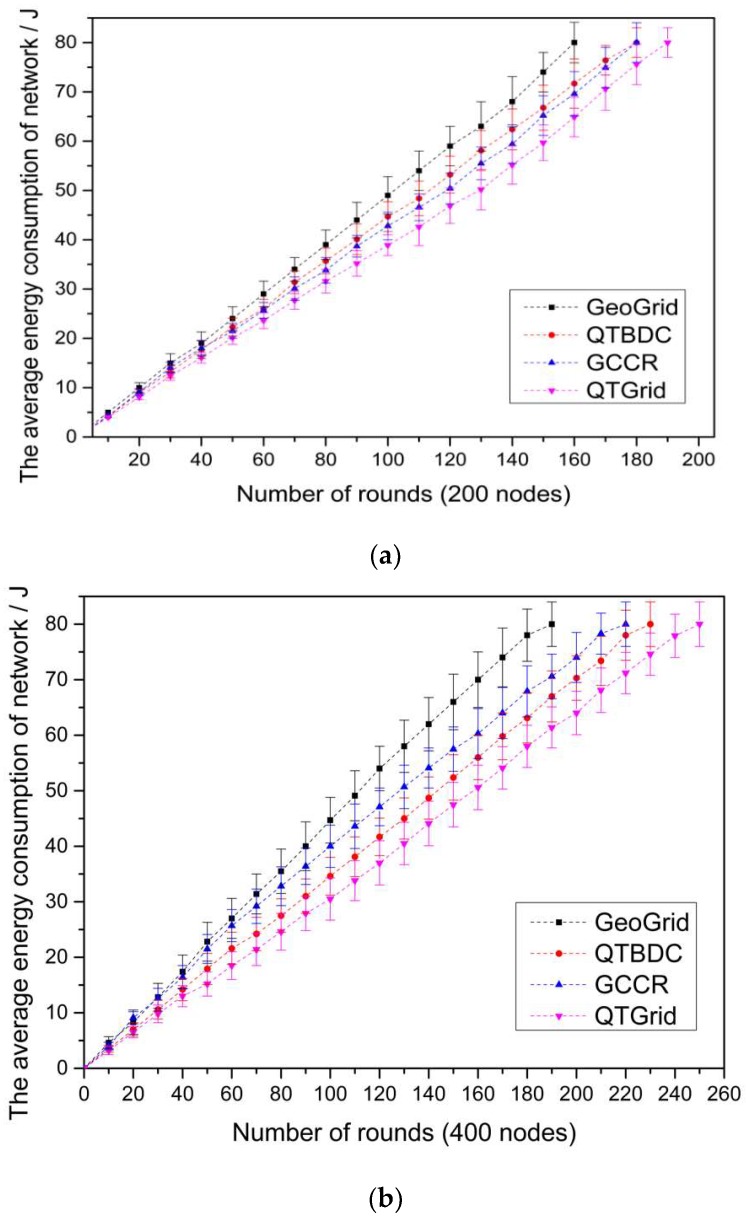
The comparison of the average network energy consumption. (**a**) Network with 200 nodes; (**b**) network with 400 nodes; (**c**) network with 800 nodes.

**Figure 9 sensors-19-02363-f009:**
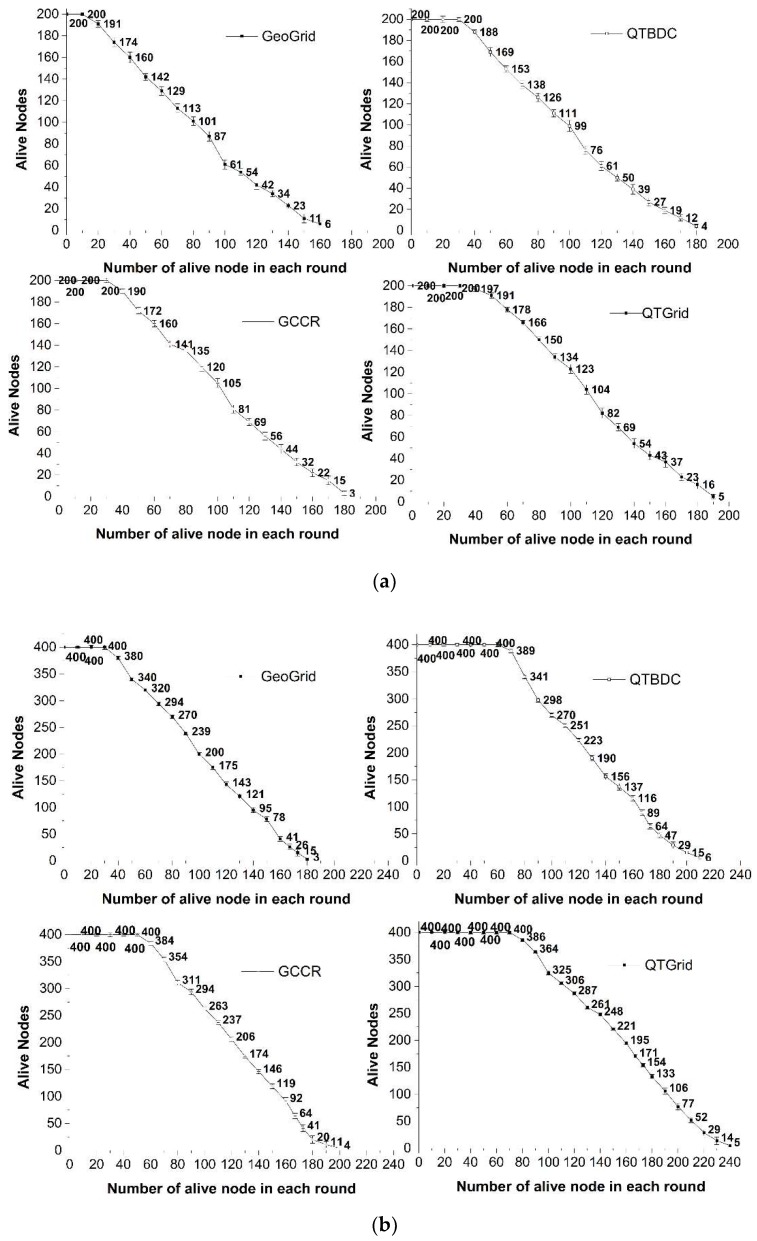
The comparison of the number of surviving nodes. (**a**) Network with 200 nodes; (**b**) network with 400 nodes; (**c**) network with 800 nodes.

**Figure 10 sensors-19-02363-f010:**
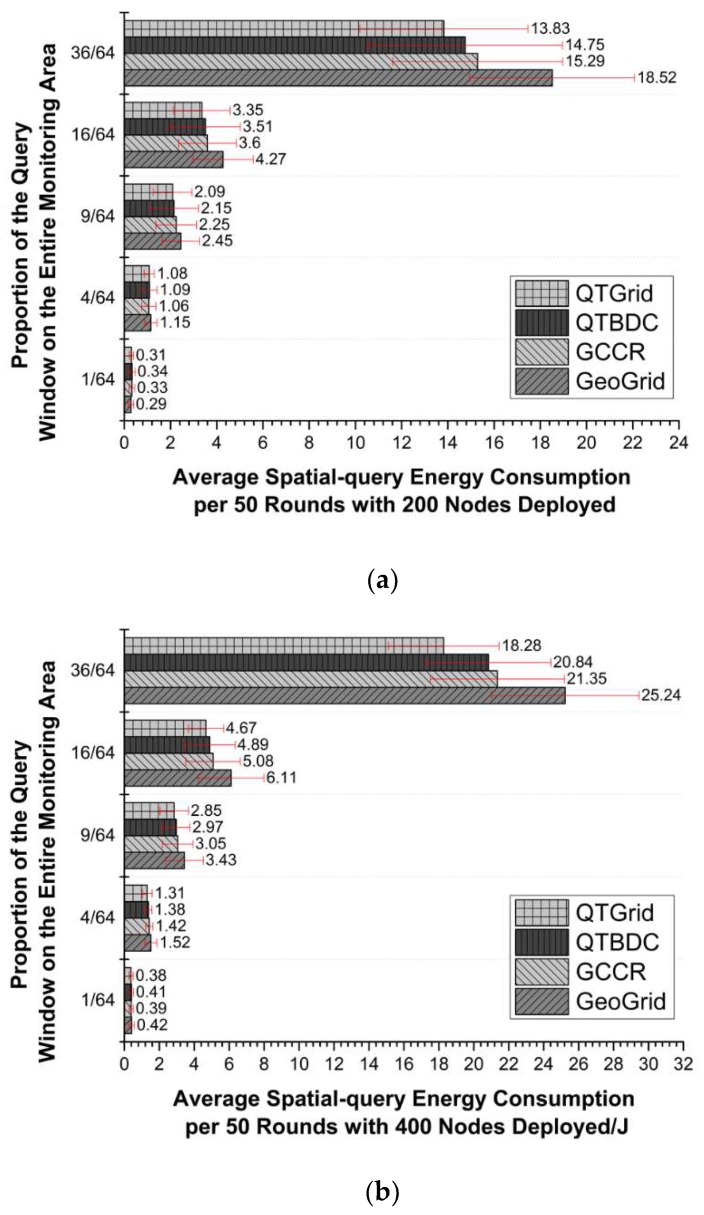
The comparison of spatial query energy consumption. (**a**) Network with 200 nodes; (**b**) network with 400 nodes; (**c**) network with 800 nodes.

**Figure 11 sensors-19-02363-f011:**
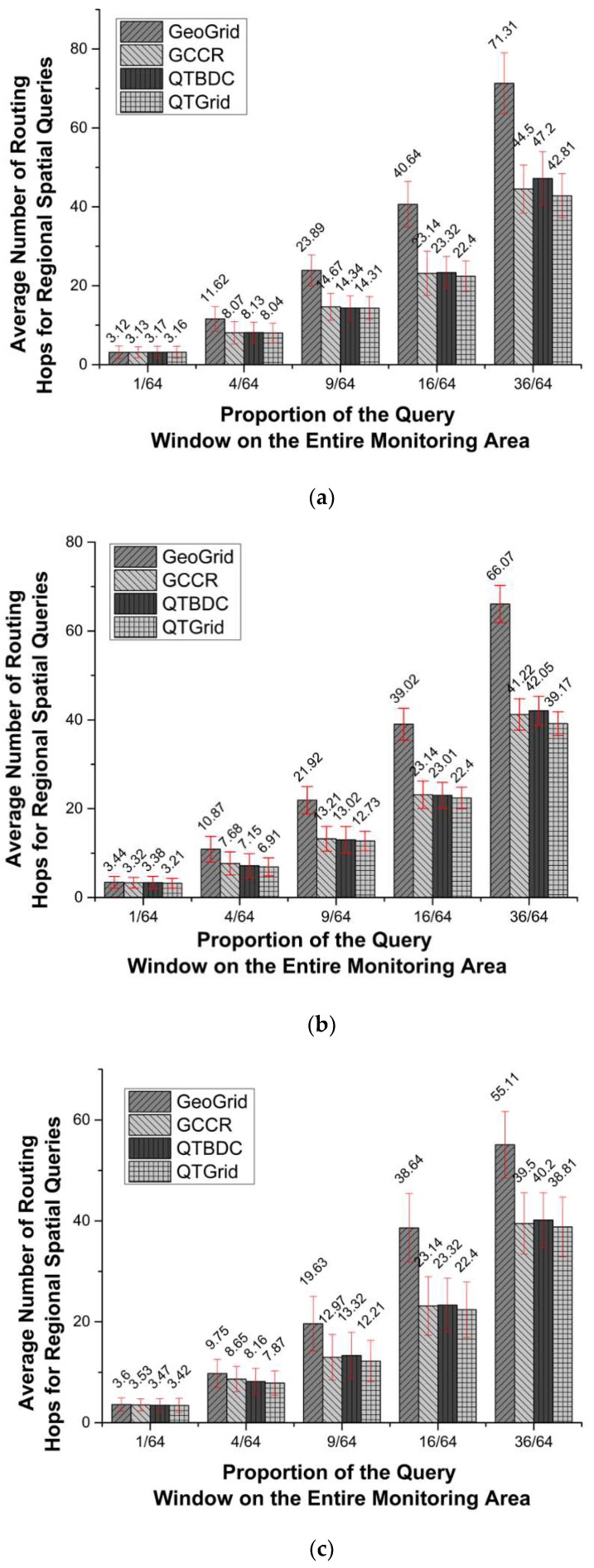
The comparison of an average number of routing hops for the spatial query. (**a**) Network with 200 nodes; (**b**) network with 400 nodes; (**c**) network with 800 nodes.

**Table 1 sensors-19-02363-t001:** Comparison of various clustering GR (Geographic routing) protocols.

Protocols	Energy Saving	Load Balancing	Spatial Query Efficiency	Large-Scale WSN
GeoGrid [26]	High	Low	Middle	Yes
GFTCRA [37]	Middle	Yes	Low	Yes
CEBCRA [16]	Middle	Yes	Middle	Yes
GCP [38]	Middle	Low	Low	Yes
EEGBR [39]	Middle	Low	Middle	No
GRCS [40]	Middle	Low	High	No
GEGR [41]	Middle	Low	Middle	No
MOCHs [42]	High	Middle	Middle	Yes
QTBDC [43]	High	Low	High	Yes
GCCR [20]	High	High	Middle	Yes

**Table 2 sensors-19-02363-t002:** Simulation parameters.

Parameters	Values
Network size (*m × m*)	160 × 160
Location of the Sink Node	Network center
Nodes’ initial energy (*J*)	10
Cluster heads’ energy threshold (*J*)	4
*E_elec_* (*nJ/bit*)	50
*ε_amp_* (*pJ/bit/m2*)	10
Computation cost of Data aggregation *E_da_* (*nJ/bit/signal*)	3
Computation cost of routing *E_r_* (*nJ/bit/signal*)	5
Number of network layers/Number of nodes	5
Radius of the intra-cluster communication (*m*)	28.28
Radius of the inter-cluster communication (*m*)	56.56
Size of the data packet (*bits*)	500
Size of the packet header (*bits*)	200

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
