# Peer review of "Energy-Efficient Spatial Query-Centric Geographic Routing Protocol in Wireless Sensor Networks"

_sensors, 2019, doi:10.3390/s19102363_

Reviewer 1 Report

The manuscript is well written, it needs to be improved only in some parts, that I am describing at last.

My main concern is about what is proposed in the manuscript and what we can find in the literature. The proposal must be clear in the manuscript, however I found an article that proposed almost the same:

M. Demirbas and X. Lu, "Distributed Quad-Tree for Spatial Querying in Wireless Sensor Networks," 2007 IEEE International Conference on Communications, Glasgow, 2007, pp. 3325-3332. doi: 10.1109/ICC.2007.551

The manuscript mention that “QTGrid was discussed from the aspects of cluster partitioning and encoding, CH election, routing strategy, and data aggregation.” Comparing the manuscript with Lu et al. 2007, the “cluster partitioning and encoding” are the same, the “CH election” is similar,  the “routing strategy” is to similar and the “data aggregation” too. The author have to describe the difference between both works in order to increase the reliability of the manuscript.

Another important concern is about the experiments. How many repetitions were made for each scenario? That is, how many network topologies did you used to plot the values on the graphs? The graphs have to present the confidence interval, which is calculated executing the same experiments several time, each one using a different network topology.

In the following there are some suggestions to improve the manuscript:

a) Lines 114 and 115 are no clear: “...joins a CH within its communication range on the basis of the cost value of the Chs”

b) Line 155: cite QTBDC [40].

c) Line 382 and 383: the definition of Erx and Ec are the same.

d) The color of the graphs are not good when you print the manuscript.

Author Response

Response to Reviewer 1 Comments

Point 1 :  The manuscript is well written, it needs to be improved only in some parts, that I am describing at last. My main concern is about what is proposed in the manuscript and what we can find in the literature. The proposal must be clear in the manuscript, however I found an article that proposed almost the same: M. Demirbas and X. Lu, "Distributed Quad-Tree for Spatial Querying in Wireless Sensor Networks," 2007 IEEE International Conference on Communications, Glasgow, 2007, pp. 3325-3332. doi:10.1109/ICC.2007.551

The manuscript mention that “QTGrid was discussed from the aspects of cluster partitioning and encoding, CH election, routing strategy, and data aggregation.” Comparing the manuscript with Lu et al. 2007, the “cluster partitioning and encoding” are the same, the “CH election” is similar, the “routing strategy” is to similar and the “data aggregation” too. The author have to describe the difference between both works in order to increase the reliability of the manuscript.

Response 1:

It is really true as reviewer suggested that the manuscript is similar to Lu’s (2007) work in some places. (However, there are still several differences between them. In the following, we compare the two protocols in terms of similarities and differences. If there is any misunderstanding of this paper, you are welcome to criticize and guide. We tried to upload the Figures but didn't work. The Figures can be viewed in the following PDF file.) Moreover, M. Demirbas and his collaborators have done a great deal of work in the study of spatial querying frameworks in WSNs (such as DQT and MDQT), which is of great help to our research. In the revised manuscript, we added some of his excellent papers, they are:

[5] Can Z, Demirbas M. A survey on in-network querying and tracking services for wireless sensor networks[J]. Ad Hoc Networks, 2013, 11(1):596-610.

[11] Demirbas M, Lu X. Distributed Quad-Tree for Spatial Querying in Wireless Sensor Networks[C]. IEEE International Conference on Communications. IEEE, 2007.

[12] Demirbas M , Lu X , Singla P . An In-Network Querying Framework for Wireless Sensor Networks[J]. IEEE Transactions on Parallel and Distributed Systems, 2009, 20(8):1202-1215.

Comparisons of DQT and our proposed   protocol

The similar   points

1.     When a   sensor device detects an event, it stores event data in its local storage and   propagates data to the upper clusterhead (CH). In this manner, data is   propagated upward in hierarchy.

2.     Selection of   PCH

The node closest to the   center of the entire network in each sub-partition is selected as the parent   node of that sub-partition. The benefit of such a selection is to avoid backward   links.

3.     The encoding   and the divition of network is similar.

4.     ……

The differences

1.     Dissemination   stage.  

In the Dissemination stage, the query is disseminated to all nodes   within the region of interest.

DQT

Our Protocol

DQT employs a type of query called Nearest Neighbor queries (NN).   It is a special case of KNN, with K = 1.

Fig. 1 illustrates a KNN query in a WSN.

Fig.1&Fig.2 are   replotted from “Rone Ilídio da Silva, Macedo D F , José Marcos S. Nogueira.   Spatial query processing in wireless sensor networks – A survey[J].   Information Fusion, 2014, 15(1):32-43”.

Windows queries are employed in our protocol.

Window queries (also called range queries) are the most common   type of spatial queries. The user defines a region of interest (called   window) and asks for data collected by the sensor nodes inside this region.   Windows are defined by rectangles in our protocol. (See page14 line 314-316)

Fig. 2 illustrates a Window query in a WSN.

2.     The   selection of CH

DQT

Our protocol

In DQT, the clusterhead at each level partition is statically   assigned to be closest node to the geographic center point of the entire   network.

In the initial stage, QTGrid selects the node closest to the   center of the grid as the CH. Then, in the re-selection of CH, QTGrid   comprehensively considers energy consumption and load balancing. The CH nodes   are selected based on metrics, such as distance from the candidate node to the   grid center and the eight adjacent CHs and residual energy.

(See page9 line 237-247)

3.     Fault   tolerance (in Lu et al. 2007)/Routing strategy (in our manuscript)

Failures may cause in   two cases are analyzed in Lu et al. 2007:

Case 1: Failures happen   before the event advertisement.

Case 2: The event has   already been published in the structure before the failure happens. Based on   these two examples, we compare the differences between DQT and our protocol.

DQT

Our protocol

In the cases of failures, to achieve resiliency while routing   to CHs or neighbors in the structure, DQT maps the DQT address of the   destination to the physical coordinates and leverages on the resilience   of a geographic routing scheme (such as GPSR) for delivering the   message. 

In case1, the event is published to the closest node (called proxy   node in paper) pretends to be the target node and finds its neighbors through   local computation by default.

In case2, queries to this node are passed to its parent node.

DQT changes the shape of a hole by changing the selection of proxy   nodes only, and has little influence on other nodes. Since GPSR only   requires single-hop information, which has already been cached as level-1   neighbors in DQT structure. When the coverage has irregular holes, the local   optimal path can be reached using right-hand rules in GPSR.

For the two cases proposed in Lu et al. 2007. Our protocol solves   the two cases from 2 aspects:

1. Since the data packet transmission in our protocol in a   grid-by-grid manner, the failure problem is concentrated on the CH of each   grid. Our protocol periodically broadcasts the CH election message to all   nodes in the network at an interval of T seconds. After receiving the   message, each cluster starts a new round of CH election. In this way, the   node with stronger capacity becomes the CHs and reduces the possibility of   nodes failure (like case1). (See page 9, section 3.2.1 and page17 line361-362   and page 22 line 416-417)

2. In order to ensure a relatively short path, our protocol selects   three CHs whose grid center are close to the Sink node as candidate next-hop   routing nodes. (Details in section 3.3, See page 12 line281-283)

Moreover, by   comprehensively considering the distance and residual energy when selecting   the next-hop routing node, our protocol QTGrid defines NHop as an evaluation   factor to select the candidate routing nodes. (See page 12, line 294)

And the routing line   are selected as follow figure 3 in our protocol.

Fig.3 Conceptual map of routing line

Point 2 : Another important concern is about the experiments. How many repetitions were made for each scenario? That is, how many network topologies did you used to plot the values on the graphs? The graphs have to present the confidence interval, which is calculated executing the same experiments several time, each one using a different network topology.

Response 2: We are very sorry for not elaborating this part in detail and we have restated in the revised manuscript.

In all scenarios, the sink node is located at the center of the network and other nodes are random deployed. For illustrating the performance of QTGrid, we compare the network lifetime and spatial query of different protocols by calculating the average value of 100 trails with different random seeds in the following Figures (9-11). Unless the sense data from the region of interest could not be transmitted to the sink node, we considered the network fail and stopped the simulation. (See page16 line 348-353)

The simulation in section 4.2: we divide the entire region into four parts with no overlap evenly, and regard the completion of spatial queries for each sub-region as a cycle. All the protocols reselect their CHs every 10 rounds. ( See page 17 line 348-352)

The simulation in section 4.3: Generating the network, we set up five different query windows, and slide the windows along the network edge until the spatial query area covers the whole network completely. A query for each window is regarded as a round. CHs are reselected every 10 rounds. (See page 21 Line 416-418)That is to say, the network topology changes with the change of cluster head node. We have re-written this part according to the Reviewer’s suggestion. (See page 21, line 397-408)

Point 3 : In the following there are some suggestions to improve the manuscript:

Thank you very much for your careful reading. We have made correction according to your comments.

a)     Lines 114 and 115 are no clear: “...joins a CH within its communication range on the basis of the cost value of the Chs”

Response 3 (a): “Second, each non-CH sensor node (also called cluster member node (CM)) selects a CH within its communication range on the basis of the cost value of the CHs”. (See page4 line 114-115)

b)     Line 155: cite QTBDC [40].

Response 3 (b): It has been changed in the revised manuscript. (See page5 line 156)

c)     Line 382 and 383: the definition of Erx and Ec are the same.

Response 3 (c): Ec is the power consumption for calculation. (See page21 line 415)

d)     The color of the graphs are not good when you print the manuscript.

Response 3 (d): Considering the Reviewer’s suggestion, we have redrawn Figures 10(a), (b), (c) and Figure 11 (a), (b), (c) in the manuscript. Compared to the previous, the new image adds texture features to make the content in the image easier to distinguish. In addition, we tried to print some of the images in black and white, which is easier to identify than before.

Special thanks to you for your good comments.

Reviewer 2 Report

This paper discusses the energy-efficient routing protocol in data-centric WSNs with spatial queries. The authors propose a new clustering geographical routing protocol called quad-tree grid (QTGrid) to save energy and improve spatial query efficiency. The authors divide the monitoring area into clusters by using a quad-tree structure and each grid is numbered by a special sequence. Then, the cluster head (CH) of each cluster is selected based on several metrics, such as distance from the candidate node to the grid center and adjacent CHs and its residual energy. Then the routing path is determined based on the residual energy of the candidate nodes and its distance to the sink node. Simulation results show the proposed protocol has lower energy consumption and higher spatial query efficiency compared with other three related protocols.

  My questions is as follows. First, the simulation results show that the proposed protocol outperforms three related protocols in terms of energy consumption. But, it can be expected since the proposed protocol is more regular than others. However, the simulation results only show the energy consumption of the proposed routing protocol. What is the performance of the proposed routing protocol as long as the transmission delay is concerned?

Author Response

Response to Reviewer 2 Comments

This paper discusses the energy-efficient routing protocol in data-centric WSNs with spatial queries. The authors propose a new clustering geographical routing protocol called quad-tree grid (QTGrid) to save energy and improve spatial query efficiency. The authors divide the monitoring area into clusters by using a quad-tree structure and each grid is numbered by a special sequence. Then, the cluster head (CH) of each cluster is selected based on several metrics, such as distance from the candidate node to the grid center and adjacent CHs and its residual energy. Then the routing path is determined based on the residual energy of the candidate nodes and its distance to the sink node. Simulation results show the proposed protocol has lower energy consumption and higher spatial query efficiency compared with other three related protocols.

Point 1: My questions is as follows. First, the simulation results show that the proposed protocol outperforms three related protocols in terms of energy consumption. But, it can be expected since the proposed protocol is more regular than others. However, the simulation results only show the energy consumption of the proposed routing protocol. What is the performance of the proposed routing protocol as long as the transmission delay is concerned?

Response 1:

We are very sorry for our incorrect writing. The network latency is shown in Section 4.3 and Figure 11and its description is incomplete, which results in it being undiscovered. The original Figure 11 is the simulation result of 200 nodes scenario. We added 400 and 800 nodes of two experimental scenarios in the revised manuscript.

We have re-written this part according to the Reviewer’s suggestion. (See page 21, line 397-411)

In section 4.3, we select energy consumption and network transmission delay as metrics for spatial query of the four routing protocols. All four protocols transmit data in a grid-by-grid manner through CHs. The size of the grid is evenly divided, and CH is generally close to the center of the grid. It can be considered that the distances between CHs is almost the same. Therefore, we define the number of transmission hops as the path of information transmission.

Usually, in a multi-hop network, the shorter the data transmission path (the fewer forwarding hops), the higher the transmission success rate [7]. In addition, the delay of data transmission is mainly caused by problems such as radio link contention, message processing time, and message queuing in per hop. Therefore, the increasing of the number of data forwarding hops has a direct impact on network latency. Meanwhile, the times of data receipt and transmission increase correspondingly, that is to say, the energy consumption is fast. Although the augment of transmission power can enlarge the communication radius of the nodes, so as to reduce the number of hops forwarded. However, this method not only increases the energy consumption of nodes, but also raise the possibility of the communication interference between neighboring nodes. In summary, we simulate the network latency by comparing the average number of routing hops.

The reason why we do not use time, mainly because the routing protocol processing time is closely related to the hardware used. Supposing these four routing protocols run on the same hardware. For hardware with strong processing power, the network delay difference is small, and for the weak processing power, the network delay difference is large.

The comparisons of QTGrid with three other protocols in terms of network latency for networks randomly distributed with 200, 400, and 800 sensor nodes (the same scenario used to evaluate the energy consumption drawn in Figure 10) are shown in Figures 11(a), 11(b), and 11(c), respectively. (See page 21&27)

Special thanks to you for your good comments.

Round  2

Reviewer 1 Report

I only miss the confidence interval in the graphs.

Author Response

Response to Reviewer 1 Comments

Point 1 :  I only miss the confidence interval in the graphs.

Response 1:

Thank you very much for reviewer's reminder.

We have added the confidence interval range with 95 percent of target values in all the Figures in Section 4 (See Line 350,351). We also corrected the color of the Figures to make the style of our manuscripts more uniform. (See Figures 9,10 and 11).

Special thanks to you for your good comments.
